# Management of Vaginal Hyperplasia in Bitches by Bühner Suture

**DOI:** 10.3390/ani12243505

**Published:** 2022-12-12

**Authors:** Roberta Bucci, Jasmine Fusi, Domenico Robbe, Maria Cristina Veronesi, Augusto Carluccio

**Affiliations:** 1Department of Veterinary Medicine, University of Teramo, Piano d’Accio, 64100 Teramo, Italy; 2Department of Veterinary Medicine, University of Milano, 26900 Lodi, Italy

**Keywords:** canine reproduction, proestral–estral disease, estrogens, vulvar suture

## Abstract

**Simple Summary:**

During the proestral–estral phase of the canine cycle, estrogens may cause exaggerated vaginal hyperplasia that can protrude through the vulvar lips. The present study aims to describe the management of vaginal hyperplasia in bitches by Bühner vulvar suture. Fourteen bitches with a complete protrusion of vaginal mucosa underwent general anesthesia for the gentle reduction of the prolapsed tissue and the application of the vulvar suture, using a Gerlach needle and a sterile vaginal suture tape; a minimal vulvar opening was maintained to allow urination. The technique proved to be useful in preventing the recurrence of the protrusion during the current estrus cycle; the Bühner suture was removed two months later, along with ovariectomy. According to the results obtained, Bühner suture is useful for the conservative treatment of vaginal hyperplasia in medium- and large-sized dogs, without any signs of trauma or ulceration of the prolapsed tissues.

**Abstract:**

Vaginal hyperplasia in bitch is an exaggerated response of the vaginal mucosa to estrogens during the proestral–estral phase of the cycle that can protrude through the vulvar lips. The present study refers to the management of vaginal hyperplasia in bitches by Bühner vulvar suture. Fourteen private-owners animals were refereed for spontaneous vaginal hyperplasia and complete protrusion of the mucosa, without ischemic or necrotic areas, which occurred during the proestral–estral phase. Under general anesthesia, prolapsed mass was cleaned with 50% glucose solution to reduce oedema and gently repositioned; the Bühner suture was applied using a Gerlach needle and a sterile vaginal suture tape, maintaining a minimal opening to allow urination in order to avoid possible recurrence in the same estrus. None of the bitches showed recurrence during the current cycle, proving the effectiveness of the Bühner suture. To prevent the possible recurrence of vaginal hyperplasia at the subsequent estrus, all the bitches underwent an ovariectomy 2 months later, when the Bühner suture was removed. In conclusion, the Bühner suture proved to be useful for the conservative treatment of vaginal hyperplasia in medium- or large-sized bitches. However, this approach should be considered only for cases in which the prolapsed mass does not show trauma, ulceration, ischemic or necrotic areas.

## 1. Introduction

In the canine species, vaginal hyperplasia is a clinical condition determined by an exaggeration of the estrogenic response of the vaginal mucosa, typically occurring during the proestral–estral phases of the canine reproductive cycle [1].

In fact, during the early stages of the estrous cycle, together with the increase in serum concentrations of estradiol, it is possible to observe the proliferation of the vaginal epithelium, up to keratinization, accompanied by marked edema of the tissues involved [2]. The increase in the volume of the mucosa sometimes is such as to cause its eversion, which can be defined as prolapse of the vaginal mucosa [3].

According to Manothaiudom and Johnston [4], vaginal hyperplasia can be didactically divided into three types, based on the degree of tissue protrusion:−Type I: slight to moderate eversion of the vaginal floor, without protrusion through the vulvar rim (Figure 1a);−Type II: protrusion, through the vulvar rim, of part of the vaginal walls or floor, to form a tongue-shaped mass with a narrow base (Figure 1b);−Type III: complete prolapse of the entire vaginal circumference, through the vulvar rim, to form a ring-shaped mass (Figure 1c).

**Figure 1 animals-12-03505-f001:**
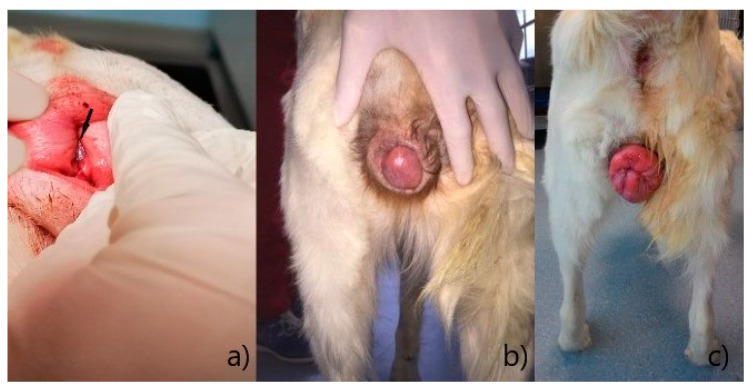
Degrees of vaginal hyperplasia [4]: (**a**) Type I, vaginal hyperplasia is indicated by an arrow; (**b**) Type II; (**c**) Type III.

Vaginal hyperplasia is described in animals from 7 months to 11 years [4], appears for the first time between the first and third estrus cycle [5], and can recur in subsequent cycles [6].

Medium- and large-sized breeds, and brachycephalic dogs seem to have a predisposition to vaginal hyperplasia [7]. The presence of a hereditary component is doubtful, although a family predisposition has been detected [1] and there are some anecdotical reports concerning a hereditary weakness of perivulvar tissues [8]. Ahuja et al. also reported a case of a dog showing vaginal protrusion one day after mating [9].

Type III prolapse is also described as an adverse effect of the induction of estrus with estradiol [10]. Moreover, a case of Type II vaginal hyperplasia has also been reported in a bitch affected by ovarian follicular cysts [11].

The protrusion resulting from vaginal hyperplasia can be complicated by the involvement of the urethra [12] and by trauma to the exposed tissues [4].

Treatment can be conservative or may require amputation of the prolapsed tract [13], followed or not by ovariectomy or ovariohysterectomy [1]. Pharmacological induction of ovulation is also described to interrupt estrus and prevent prolapse recurrence during the same estrus cycle [9].

The aim of the present paper is to describe a conservative surgical treatment for Type III vaginal hyperplasia that involves the reduction of eversion and the application, to prevent a recurrence, of the Bühner vulvar suture, a commonly used technique in buiatrics, for the treatment of vaginal prolapse in cows [14] and ewes [15]; however, it is also described in a modified form for the bitch [6,9,16].

## 2. Materials and Methods

The clinical study was developed at the Veterinary Teaching Hospital of the University of Teramo and involved 14 bitches referred for the presence of vaginal hyperplasia with a protrusion, beyond the vulvar lips, of the entire circumference of the vaginal mucosa. For each patient, the owners signed informed consent for medical and surgical treatment.

Each subject underwent a general evaluation and a clinical examination of the reproductive system to verify the degree of prolapse, the integrity of the prolapsed tissues, and any involvement of the urethra and the bladder.

Vaginal cytology was also performed to determine the stage of the estrous cycle, and an ultrasound examination was performed to rule out ovarian and uterine abnormalities.

To reduce the vaginal prolapse, all the patients underwent general anesthesia, following a basic blood count and biochemical test.

The anesthetic protocol provided for intramuscular premedication with 0.2 mg/kg of methadone (Semfortan^®®^, Eurovet Animal Health B.V, Bladel, The Netherlands) and 5 µg/kg of dexmedetomidine (Dexdomitor^®®^, Vétoquinol, Cusago, Italy), followed by intravenous induction with propofol (Proposure^®®^, Boehringer Ingelheim Animal Health Italia, Noventana, Italy) and maintenance with isoflurane (Isoflo^®®^, Zoetis Italia, Roma, Italy).

For intraoperative pain control, a continuous infusion of 0.5 µg/kg/h dexmedetomidine and 2 µg/kg fentanyl infusion (Fentadon^®®^ Eurovet Animal Health B.V. The Netherlands) were associated when necessary. Anti-inflammatory treatment with 0.2 mg/kg of meloxicam has also been started (Metacam^®®^, Boehringer Ingelheim Italy).

Patients were positioned in sternal recumbency, with the hindlimbs off the table [17], inserting a soft thickness below the pelvis, to prevent nerve compression [18]. The preparation of the patients also included a wide trichotomy of the anogenital area, as well as part of the hind limbs and wrapping of the tail (Figure 2).

The prolapsed tissues were cleansed with saline and sprayed abundantly with 50% glucose solution (Glucose S.A.L.F. 50% ^®®^, S.A.L.F., Cenate Sotto, Italy) to reduce edema. After a thorough surgical scrub, manual repositioning of the eversion was carried out: starting from the less exposed section, a gentle retropulsion of the vaginal walls was performed, simultaneously applying a gentle compression to facilitate the repositioning of the prolapsed tissues in a physiological condition. A second operator maintained constant pressure to hold repositioned tissues in place. Finally, digital palpation of the vagina ensured complete distension of the tissues.

To prevent a recurrence, a suture was performed on the vulva with the Bühner technique, as described by Matteuzzi [16]. After a subcutaneous injection with 0.5 mL lidocaine (Lidocaine 2% ^®®^ Ecuphar Italia, Milano, Italy) diluted 1:1 with saline solution, an incision of 0.5 cm was made at 1 cm distance from the dorsal commissure and one at the same distance from the ventral commissure; a 20 cm Gerlack needle (Figure 3a) was therefore inserted into the ventral incision and pushed through the subcutaneous tissue (Figure 4a), laterally to the vulva in order to exit at the dorsal incision; one end of a 6 mm sterile tape (vaginal tape for cattle, Kruuse, Langeskov, Denmark) (Figure 3b) was inserted into the eyelet on the Gerlack needle and pulled back through the subcutaneous tissue to exit ventrally (Figure 4b); the same operation was repeated with the opposite side of the vulva, using the free dorsal end of the vaginal bandage (Figure 5a).

Therefore, in the subcutaneous tissue around the vulva, a lace was created and tied under the ventral commissure to reduce the vulvar opening and prevent new protrusion of the vaginal tissues, but allowing the patient to urinate normally (Figure 5b). The incisions were closed with a detached suture (2/0 Vicryl^®®^ Ethicon US, Raritan, NJ, USA). The knot of the vaginal tape was included in the suture of the ventral incision to prevent accidental openings and contamination.

After surgery, the patients were hospitalized for one week. All the animals underwent systemic anti-inflammatory therapy with 0.1 mg/kg meloxicam for five days and antibiotic therapy with 5 mg/kg enrofloxacin (Xeden^®®^, Ceva Salute Animale S.p.A., Agrate Brianza, Italy) for one week. Surgical wounds were regularly cleaned and the lapping was prevented by applying the Elizabethan collar. The occurrence of a new protrusion, and urinary or fecal tenesmus was also assessed daily during hospitalization. To follow the trend of the estrous cycle, the patients were subjected to colpo-cytological examination every 3 days, until the onset of cytological diestrus.

After two months, with the informed consent of the owners, all the bitches underwent ovariectomy. At the same time, the Bühner suture was removed, incising the perivulvar tissue again and gently removing the bandage after cutting it.

## 3. Results

The subjects involved were from 1 to 7 years of age and presented in good clinical condition. They all belonged to medium- and large-sized breeds, weighing between 20 and 46 kg, in line with the standards of the respective breeds (Table 1).

The most represented breeds were the Retrievers, the Dogo Argentine, and the Abruzzese Shepherds.

In all the patients, the gynecological examination revealed a rounded mass with a central lumen protruding from the vulvar lips. Four subjects presented small abrasions of a few millimeters on the tissues involved.

In a 1-year-old Abruzzese Shepherd dog, the urethral meatus was also involved in the eversion of the vaginal mucosa (Figure 6), and a urinary catheter was inserted to allow the bladder to empty and facilitate subsequent repositioning of the prolapsed tissues.

For all the subjects included, the vaginal smear examination confirmed the proestral/estral phase (predominance of superficial and cornified cells) [19], while the ultrasound examination excluded uterine abnormalities. The blood count and biochemical tests were within physiological ranges for the species.

No urinary difficulties or fecal tenesmus were reported for any subject.

In four subjects, between the second and fourth day after suture application, a recrudescence of vaginal hyperplasia was detected, without eversion through the vulvar lips (Figure 7): a gentle digital palpation of the vagina allowed a new reposition of the tissues, without the need for further surgical interventions. In these subjects, the vaginal smear showed a cytology compatible with estrus (over 90% of cornified superficial cells) [19]. Vulvar suture with Bühner technique proved to be effective in preventing a new prolapse; the residual opening of the vulva was, in any case, enough to allow normal urination and the application of appropriate gynecological examinations.

Overall, for all the subjects, the evaluation of vaginal smears showed the onset of cytological diestrus (intermediate and parabasal cells) [19] in 7–10 days from the onset of prolapse.

After the removal of Bühner’s tape, no adhesions or strictures of the vulva were detected.

## 4. Discussion

Vaginal hyperplasia is a clinical condition that typically occurs during the follicular phase of the canine estrus cycle [5].

Some authors distinguish a “true” vaginal prolapse from vaginal protrusion, also known as vaginal fold prolapse [6,20]: the former occurs around parturition, when the concentration of serum progesterone declines and the concentration of serum estrogen increases [3,8]; the latter is, instead, a consequence of vaginal hyperplasia, typically occurring during the proestral–estral phase of the estrus cycle [1,3]. The difference between the two cases is the involvement of the entire vaginal wall in the case of “true” prolapse (and, in some cases, also the bladder, cervix, and colon) [21], while following vaginal hyperplasia, the protrusion only affects the vaginal mucosa [3].

True vaginal prolapse is described both in bitches in the last third of gestation [22,23] and in animals at the second stage of delivery: in this case, the prolapse can be the cause of dystocia [8] or, in milder cases, can allow puppies to pass normally through the vaginal canal [22]. Prassinos et al. [24] reported a case complicated by the rupture of the prolapsed vaginal wall following parturition, with consequent herniation of the uterus, bladder, and intestinal loops.

Prolapse following vaginal hyperplasia, instead, is frequently seen in young females within the third estrus cycle [7]. In the present study, the age of the patients referred for vaginal hyperplasia ranged from 1 to 7 years, in agreement with Manothaiudom and Johnston [4], who report a range of 7 months to 11 years for clinical presentation. The wide age range is justified by the fact that, during several estrous cycles, vaginal hyperplasia can cause protrusions of varying severity, ranging from type I to type III, in the same subject.

The size of the included bitches is also in agreement with the typical signaling of patients affected by this pathology: several authors report a prevalence of vaginal hyperplasia in large breeds [6,9,10]. As for the breeds involved, the consulted literature also cites Mastiffs, Dalmatians, and Dobermanns [25]. Furthermore, many papers report a certain predisposition in brachycephalic dogs, particularly the Boxer [1,16]. In the authors’ experience, even the herd guardian dogs (Abruzzese Shepherd dogs and similar) would seem to be predisposed, as well as Retriever dogs.

In the canine species, vaginal mucosal prolapse is considered an emergency when the involved tissue is dry, necrotic, or traumatized and when the mass prevents urination [4]. In the cases considered, the tissues involved were in good condition, not presenting necrosis or ulcers; only in one case, the involvement of the urethral meatus prevented correct urination and it was necessary to apply a urinary catheter. Post et al. [12] reported a severe case of vaginal hyperplasia, in which the bladder was also involved in the prolapse, making it difficult to apply the urethral catheter.

Various therapeutic approaches are described in the literature. Several authors [1,25] report conservative treatments based on cleaning, hydration, and prevention from trauma of prolapsed tissues, associated or not with pharmacological protocols to induce ovulation. Moreover, surgical resolution involving the amputation of the prolapsed tract [1,11] is described, especially in cases complicated by ulcerations and necrosis.

In the cases considered, a conservative approach was chosen as the vaginal mucosa did not present ischemic or necrotic areas, suggesting a good recovery after adequate therapy. The bibliography consulted supports this choice, as several cases of Type III vaginal mucosal prolapse are reported and treated with vulvar repositioning and suturing [6,9,16,20].

Despite all the treated patients being in good general condition and the prolapsed mucous membranes not presenting lesions, the authors preferred to proceed with repositioning the prolapse and the vulvar suture. The choice was made to prevent the risk of self-trauma or infections in the following days. The evolution of the estrus cycle was therefore monitored, waiting for ovulation and subsequent diestrus. During this estral phase, the hormonal profile changes: the drastic decrease in serum estrogen levels and the increase in progesterone favor the reduction of edema, and the spontaneous resolution of hyperplasia [2].

Regarding the anesthetic protocol, Post et al. in 1991 [1] stated that general anesthesia of the patients was not necessary for surgical resolution with amputation; however, mild sedation was enough. More recent communications, on the other hand, always report interventions on patients under general anesthesia, also integrated with lumbosacral locoregional [20] for adequate intraoperative pain control, respecting animal welfare. Differently from the sources cited, in the cases under examination, intraoperative pain was managed systemically to modulate it based on the clinical responses detected. Local infiltration of lidocaine for the application of the Bühner suture is not reported in similar cases for the canine species; however, it is used in buiatrics [26] in association with epidural block [27].

In preparation for repositioning, in the cases under examination, it was chosen to pour on the prolapsed tissues with a 50% glucose solution to partially reduce edema and facilitate corrective manual procedures. For this purpose, the use of 50% Dextran [24], but also of potassium permanganate [9], is reported in the literature. This precaution is borrowed from the history of buiatrics, when the prolapsed viscera were sprinkled with sugar to reduce edema, as also reported by Sarrafzadeh-Rezaei and collaborators [10], for use on a bitch with vaginal prolapse resulting from the induction of the estrus.

The manual repositioning performed is in line with what was described by Ahuja et al. [9]. The repositioning method is not specified for the other sources consulted. The aim of this procedure, however, must always be to bring the tissues back to their physiological position after a thorough cleansing, being very careful to the urethral meatus and avoiding that the procedure itself can damage the mucosa.

Vaginal suturing using the Bühner technique was first described in 1960 to prevent the recurrence of vaginal prolapse in cows. In ruminants, unlike the canine species, vaginal prolapse is a common pathology in pregnant animals, mostly in the third trimester of gestation [26]. However, the use of this technique has also been described for the equine species [28] and for the canine species [16].

In the present paper, the Bühner suture was applied with the classical method, as described by Matteuzzi for the canine species [16], because the size of the patient allowed easy use of the Gerlack needle and the vaginal suture tape. However, both Zedda et al. [6] and Ahuja et al. [9] described a modified version for the bitch, which involves creating a buttonhole around the vulva using a normal suture. Other authors [20,22] have performed vaginal sutures for the prevention of recurrence, without specifying the technique. Regardless of the technique and materials used, the vaginal suture must reduce the vulvar opening just enough to prevent a new prolapse of the hyperemic tissues; however, it must at the same time allow the bitch to urinate normally.

In four of the treated bitches, a Type I vaginal hyperplasia was still found in the days following the surgery; however, the suture applied prevented a new prolapse. This situation was determined by the prolongation of the follicular phase, as confirmed by the control vaginal smears. To prevent this, the bibliography also reports the use of pharmacological protocols to induce ovulation [1,20]; however, the authors chose not to pharmacologically manipulate the estrous cycle, being able to manage the patients instead, waiting for the physiological diestrus.

Due to the aforementioned size and breed predispositions, and the correlation between vaginal hyperplasia and hormonal profile during the estrus cycle of bitches, vaginal prolapse may recur [6]. To prevent this occurrence, the bibliography reports ovariectomy or ovariohysterectomy as resolutive methods for bitches not intended for breeding [4].

The authors preferred to solve the prolapse by repositioning and applying the Bühner suture, instead of performing an ovariectomy directly, as this procedure is less invasive and easier to perform, especially if performed urgently to restore the prolapsed tissues, and avoid trauma and necrosis of the vaginal mucosa. In addition, ovariectomy is a surgical procedure that can lead to complications [29]; moreover, in the authors’ opinion, it is preferable to perform it as a planned intervention and not in emergency.

On the other hand, spaying prevents the risk of recurrence; therefore, all the animals involved in the study, not intended for reproduction, underwent an ovariectomy. This procedure was performed two months later, at the end of the diestrus, to avoid the risk of clinical signs of pseudopregnancy [30].

The vulvar suture was removed at the same time as sterilization as, despite being a minimally invasive procedure, it requires mild sedation of the patient; this choice was made, in respect of animal welfare, to reduce anesthetic procedures to those strictly necessary. In any case, the vaginal tape can be removed as soon as vaginal cytology or progesterone assays show diestrus, as recurrences are less likely at this stage.

## 5. Conclusions

In conclusion, the data obtained demonstrate that the vulvar suture with Bühner technique is effective in preventing the recurrence of prolapse in bitches with vaginal hyperplasia.

However, the execution of this technique with the classical method, as applied in buiatrics, is advisable only in medium- to large-sized bitches and when the prolapsed tissues are in good condition, without ulcerations, ischemic or necrotic areas.

## Figures and Tables

**Figure 2 animals-12-03505-f002:**
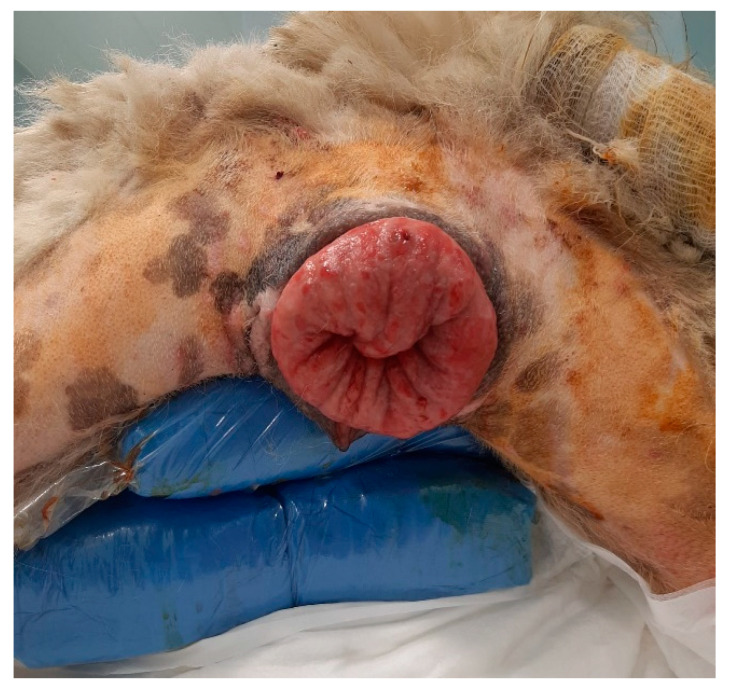
Patients are positioned in sternal recumbency, with the hindlimbs off the table and a soft thickness below the pelvis. The tail is wrapped and hairs are clipped from the tail to the limbs.

**Figure 3 animals-12-03505-f003:**
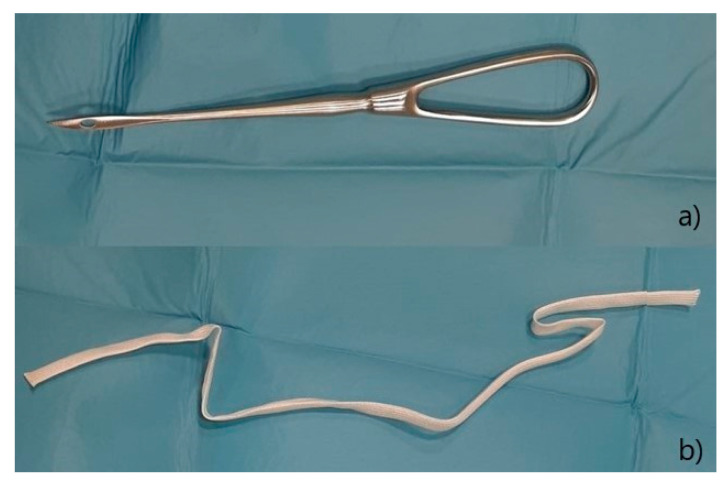
(**a**) Gerlach needle; (**b**) sterile vaginal tape.

**Figure 4 animals-12-03505-f004:**
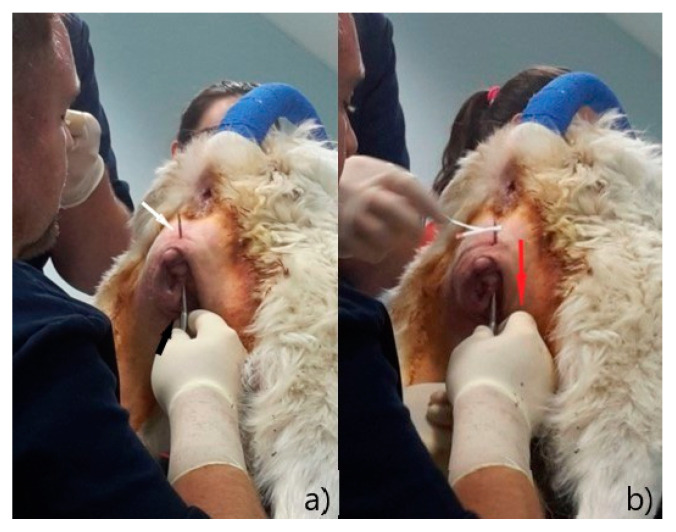
Bühner technique: (**a**) place the Gerlach needle into the ventral incision (black arrow) and push it through the subcutaneous tissue, laterally to the vulva, exiting at the dorsal incision (white arrow); (**b**) thread one end of the tape into the eyelet of the needle and pull it back, through the subcutaneous tissue (red arrow).

**Figure 5 animals-12-03505-f005:**
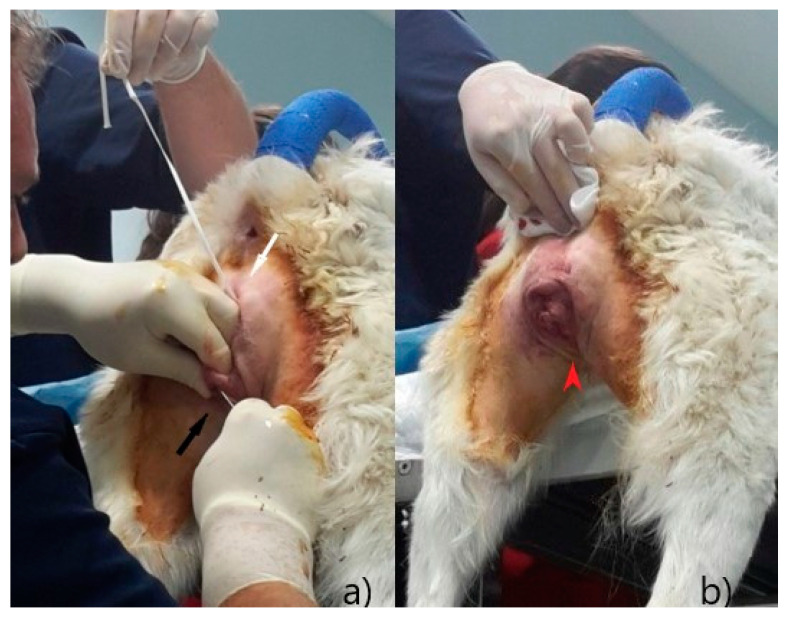
Bühner technique: (**a**) place the Gerlach needle into the ventral incision (black arrow) and push it through the subcutaneous tissue, laterally to the vulva, exiting at the dorsal incision (white arrow); thread the dorsal free end of the tape into the eyelet of the needle and pull it back, through the subcutaneous tissue so that the tape loops surrounding vulvar labia; (**b**) tie a knot with the vaginal tape exiting the ventral incision (red arrow); reduce the vulvar opening up to allowing proper urination.

**Figure 6 animals-12-03505-f006:**
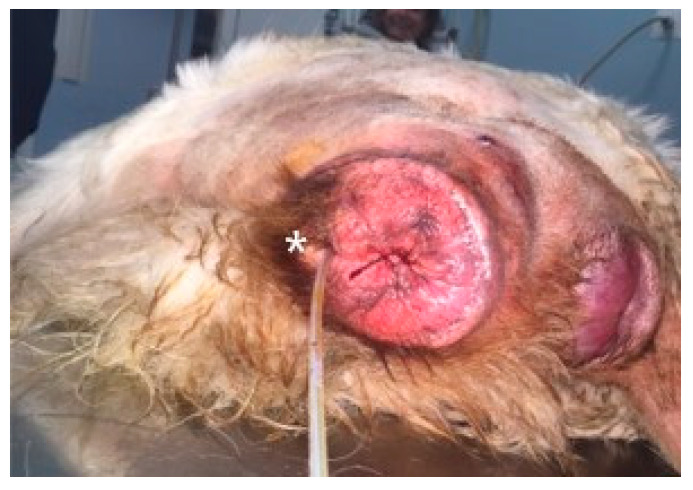
Type III vaginal hyperplasia, with urethral involvement (*) in the prolapsed mass. Urethral catheterization was performed prior to prolapse reduction and vulvar suturing.

**Figure 7 animals-12-03505-f007:**
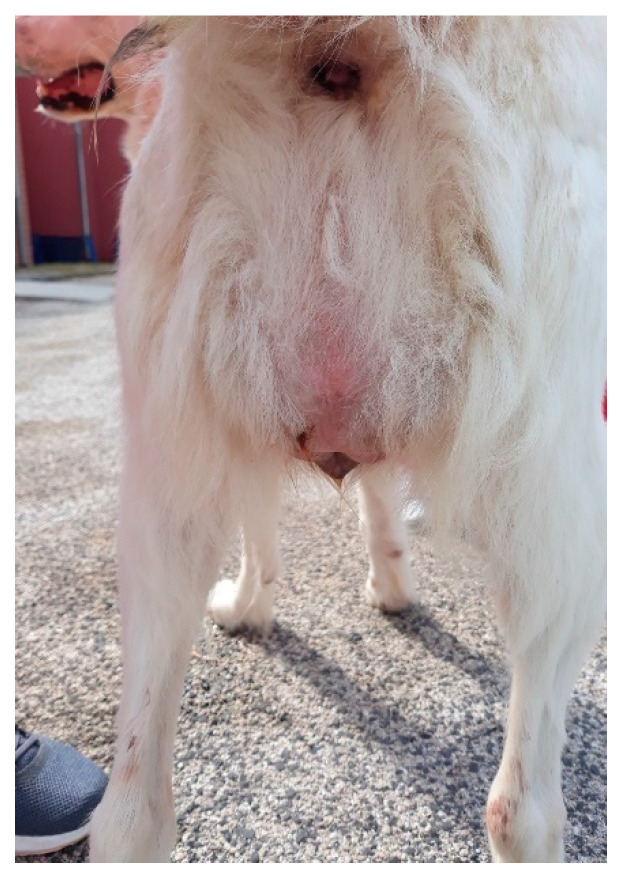
Recrudescence of vaginal hyperplasia. Bühner suture proved to be useful for the prevention of a new prolapse through the vulvar lips.

**Table 1 animals-12-03505-t001:** Signalment and history of animal involved in the study.

Patient	Breed	Age (Years)	Weight (kg)	Clinical Presentation
1	Golder Retriever	3	28	Type III with small abrasion
2	Mixed	2	27	Type III
3	Mixed	4	29	Type III with small abrasion
4	Abruzzese Shepherd	1	31	Type III with urethral involvement and abrasion
5	Golden Retriever	5	27	Type III
6	Italian Mastiff	2	41	Type III
7	Labrador Retriever	2	24	Type III
8	Mastiff	1	46	Type III
9	Dogo Argentino	2	37	Type III
10	Cocker Spaniel	7	20	Type III
11	Italian Bracco	2	31	Type III
12	Dogo Argentino	1	39	Type III with small abrasion
13	Golden Retriever	2	30	Type III
14	Abruzzese Shepherd	1	36	Type III

## Data Availability

The data presented in this study are available on request from the corresponding author.

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
