# Peer review of "Management of Vaginal Hyperplasia in Bitches by Bühner Suture"

_animals, 2022, doi:10.3390/ani12243505_

Round 1

Reviewer 1 Report

Figure 1a could be improved upon. Less fingers, gloved fingers, an arrow indicating the hyperplasia. 

Lines 62 and 65, regarding references 8 and 9, need mention of validity. Or remove, not contributory and likely anecdotal.

Line 90 should be estrus? Or confirm stage of estrous cycle is estrus? Line 91 include evaluation of ovaries as well. Line147 why was enrofloxacin chosen? Big gun and not really an optimal choice for skin. 

Figure 4 needs clarification, perhaps a graphic image (drawn) would help, also place arrows to help description. I am unclear as to exactly how to place the needle. Figure 5 also needs labels. Is there a better image of the completed suture showing where the knot is? 

Author Response

Reviewer 1

  • Figure 1a could be improved upon. Less fingers, gloved fingers, an arrow indicating the hyperplasia. 

Thank you for your suggestions, figure 1a has been improved, and footnotes have been corrected, as well. (Lines 56-57)

  • Lines 62 and 65, regarding references 8 and 9, need mention of validity. Or remove, not contributory and likely anecdotal.

The manuscript has been modified as follow “.., and there are some anecdotical reports concerning a hereditary weakness of perivulvar tissues [8]. Ahuja et al. also reported a case of a dog showing vaginal protrusion one day after mating [9]” to highlight that these conditions are from anecdotical reports and there are not validated, yet. (Lines 63-65)

  • Line 90 should be estrus? Or confirm stage of estrous cycle is estrus? Line 91 include evaluation of ovaries as well.

The cytology was performed to determine the stage of the estrous cycle. Results from vaginal smears are reported in lines 182-183 and confirm that all bitches were in estral stage of the cycle. Yes, ovarian were included as well, I apologize for the forgetfulness.

The manuscript has been modified as follow: “Vaginal cytology was also performed to determine the stage of the estrous cycle, and an ultrasound examination was performed to rule out ovarian and uterine abnormalities”. (Lines 90-91)

  • Line147 why was enrofloxacin chosen? Big gun and not really an optimal choice for skin. 

The choice of antibiotic was not made only for the prevention of infections at the skin incision sites, that have been treated also with local medicaments, but also in order to prevent uterine ascendent infections. In fact, surgical procedures were performed in a sterile way, but, in the Authors opinion, there was a risk of previous contamination of the prolapsed tissue.

Fluoroquinolones were preferred for their bioavailability in the reproductive tract, but also basing on previous report on their effectiveness in cases of pyometra (Ros, 2014; Melandri, 2019).

  • Figure 4 needs clarification, perhaps a graphic image (drawn) would help, also place arrows to help description. I am unclear as to exactly how to place the needle.

Thank you for the suggestions. Figure 4 has been divided into figure 4ab and 5ab. Marks and arrows have been added to better identify where to insert the needle. Descriptions have been implemented as well. (Lines 140-152)

All reference to the figures in the text have been corrected.

  • Figure 5 also needs labels.

Figure 5 (now figure 6) has been modified. An asterisk has been added to identify the urethra. (Lines 180-181)

  • Is there a better image of the completed suture showing where the knot is?

The knot is completely covered by the skin, when suturing the ventral skin incision. A red arrow has been added in figure 5b to identify the position of the knot.

A sentence has been added in the manuscript, as well, to better explain this step (lines 137-138)

The knot of the vaginal tape was included in the suture of the ventral incision to prevent accidental openings and contamination.”

Reviewer 2 Report

Dear Authors, 

This manuscript of yours on canine vaginal hyperplasia management is interesting and add value to the practice. No ethical approval is mentioned. You should either provide that or clearly define that this is a case series. Furthermore, my suggestion is to have a native English speaker revising the manuscript. 

A few minor suggestions:

- Choose whether to use 'Type 3' or 'Type III' (e.g., lines 65 and 75)

- Why the bitches were hospitalized for one week?

- Is it applicable in smaller dogs?

- Line 165 'Four' instead of 'four'.

- Line 174: 'keratinized' should be changed in 'superficial cells' or 'cornified superficial cells'. I guess they might be in proestrus as well: change in proestrus/estrus or describe the cytology thoroughly.

Author Response

Reviewer 2

Dear Authors, This manuscript of yours on canine vaginal hyperplasia management is interesting and add value to the practice. No ethical approval is mentioned. You should either provide that or clearly define that this is a case series. Furthermore, my suggestion is to have a native English speaker revising the manuscript. 

Thank you for your suggestions: concerning ethical approval, the following sentence has been added in the proper section: “This research does not fall within Directive 63/2010 of the European Parliament and of the Council on the protection of animals used for scientific purposes (transposed into Italian law by Legislative Decree 26/2014) and thus doesn’t require any authorization from the national competent Authorities”. An appropriate letter of exemption from the Ethical Committee of Teramo University has been addressed to the Editor.

Moreover, the manuscript has been carefully checked by a native English colleague

A few minor suggestions:

  • Choose whether to use 'Type 3' or 'Type III' (e.g., lines 65 and 75)

The spelling has been corrected (Type III) in line 76, thank you.

  • Why the bitches were hospitalized for one week?

Patients were hospitalized for a better control after surgery, to promptly identify any recrudescence or problems in urination and defecation, as well as for the correct administration of therapies. Hospitalization was not strictly necessary, but agreed with the owners, for an accurate follow up. All the patients were hospitalized in adequately sized boxes, regularly fed three times a day and regularly taken out for walks by both the veterinary staff and nurses and the owners during the visits.

  • Is it applicable in smaller dogs?

The limitation of this procedures is the size of Gerlach needle currently available (15 cm the smaller). In Authors’ experience, the procedure would be applicable in dogs from 10 to 20 kg, but the size of the needle makes it less easy to perform. In smaller dogs, the modified technique, as described by Professor Zedda (2016), would be preferable.

  • Line 165 'Four' instead of 'four'.

Corrected, thank you.

  • Line 174: 'keratinized' should be changed in 'superficial cells' or 'cornified superficial cells'. I guess they might be in proestrus as well: change in proestrus/estrus or describe the cytology thoroughly.

The manuscript has been modified as follow: “For all subjects included, the vaginal smear examination confirmed the proestral/estral phase (predominance of superficial and cornified cells) [19]”. (Lines 182-183)